Saliency detection of textured 3D models based on multi-view information and texel descriptor

Zhang Ya
Chen Chunyi chenchunyi@hotmail.com
Hu Xiaojuan
Li Ling
Li Hailan
School of Computer Science and Technology, Changchun University of Science and Technology , Changchun , Jilin , China
Jhanjhi N
Electronic publication date: 2023 Oct 25
Publication date: 2023
Volume: 9
Electronic Location ID: e1584
Received 2023 Mar 31; Accepted 2023 Aug 20
Copyright: ©2023 Zhang et al.
Copyright year: 2023
Copyright holder: Zhang et al.
License: This is an open access article distributed under the terms of the Creative Commons Attribution License, which permits unrestricted use, distribution, reproduction and adaptation in any medium and for any purpose provided that it is properly attributed. For attribution, the original author(s), title, publication source (PeerJ Computer Science) and either DOI or URL of the article must be cited.
License URL: https://creativecommons.org/licenses/by/4.0/

Keywords: Image saliency, Mesh saliency, Textured model saliency, Multi-view, Computer graphics, Perception, Region of interest, Dataset, Computer vision, Human eye fixation

Funding: The National Natural Science Foundation of China U19A2063 The Jilin Provincial Science & Technology Development Program of China 20230201080GX This work was supported by the National Natural Science Foundation of China under Grant U19A2063 and by the Jilin Provincial Science & Technology Development Program of China under Grant 20230201080GX. There was no additional external funding received for this study. The funders had no role in study design, data collection and analysis, decision to publish, or preparation of the manuscript.

==============================
Saliency-driven mesh simplification methods have shown promising results in maintaining visual detail, but effective simplification requires accurate 3D saliency maps. The conventional mesh saliency detection method may not capture salient regions in 3D models with texture. To address this issue, we propose a novel saliency detection method that fuses saliency maps from multi-view projections of textured models. Specifically, we introduce a texel descriptor that combines local convexity and chromatic aberration to capture texel saliency at multiple scales. Furthermore, we created a novel dataset that reflects human eye fixation patterns on textured models, which serves as an objective evaluation metric. Our experimental results demonstrate that our saliency-driven method outperforms existing approaches on several evaluation metrics. Our method source code can be accessed at https://github.com/bkballoon/mvsm-fusion and the dataset can be accessed at 10.5281/zenodo.8131602.

Introduction

Many graphic applications, including mesh quality assessment (Abouelaziz et al., 2020) and viewpoint selection (Abid, Perreira Da Silva & Le Callet, 2019), have benefited from research on mesh saliency. This concept was first proposed by Lee, Varshney & Jacobs (2005), who mainly used geometric measures such as curvature. However, given that texture also plays a crucial role in human visual perception, we argue that geometry-based metrics may not adequately capture salient regions of a textured model. This point is illustrated by the clear difference between Figs. 1B and 1C. Some researchers have recognized this issue and proposed innovative methods that incorporate texture cues, such as Yang et al. (2016). Despite their promising results, these methods often have limitations due to a lack of datasets.

Figure 1 Here is a bunny model with several saliency maps displayed as heatmaps.

(A) The source textured model. (B) The saliency map of human eye fixation collected by our experiment. (C) The saliency map provided by 3DVA dataset (Lavoué et al., 2018) which reflects human eye fixation on pure mesh. (D) The saliency map calculated by our method on textured model. Warmer color represents more salient.

In 3D data simplification, visually salient regions are often oversimplified due to their low geometric importance. To address this issue, recent studies have successfully employed neural networks to capture salient parts accurately. Noteworthy achievements have been reported in works such as Nousias et al. (2020) and Song et al. (2021). In the case of textured models, some studies have proposed methods for generating saliency maps by combining geometric and texel features. For example, Yang et al. (2016) proposed a non-linear normalization-based approach while Nouri, Charrier & Lézoray (2017) introduced Global Visual Saliency built on the Dempster-Shafer theory. Similarly, several techniques have been developed in point-cloud analysis, such as Zheng et al. (2019) and Tinchev, Penate-Sanchez & Fallon (2021), with the goal of identifying visually significant regions based on their local geometric properties. Inspired by those works, to better reflect the information perceived by the human eye, we introduce a method that combines multi-view projection information of a 3D textured mesh with a texel descriptor. Our approach, called Multi-View-Saliency-Map Fusion (MVSM-Fusion), integrates both aspects effectively, leading to accurate saliency maps. The first part of MVSM-Fusion, the fusion of multi-view information, is derived from the intuitive idea that 2D saliency maps in each view image represent the projection of one optimal 3D saliency map onto 2D space. Given that the current local descriptor provides a precise representation of local geometric properties, it lacks texture considerations. Therefore, we designed a novel descriptor, which forms the foundation of another part of MVSM-Fusion and incorporates texture.

The dataset proposed by Lavoué et al. (2018) does not fully reflect human eye fixations on textured models. To address this limitation, we conducted an experiment to record human gaze on textured models. Our dataset was constructed using seven observers and a variety of models obtained from Zhou et al. (2005) and online resources. To evaluate 3D saliency detection methods, Song et al. (2021) recommended using the area under the ROC curve (AUC). However, in this study, we evaluated our proposed method by computing the similarity between the 2D ground truth and the multi-view saliency output. Results show that our method outperforms the existing method in terms of Pearson’s correlation coefficient (CC), similarity metric (SIM), earth mover’s distance (EMD), and normalized scanpath saliency (NSS).

Overall, our work provides valuable resources and insights for advancing research on saliency modeling in 3D textured scenes. Our contribution can be summarized in two parts:

1) We have created an open dataset of 3D textured models annotated with human eye fixations, which can be used to evaluate saliency prediction methods’ accuracy.

2) We have developed a novel saliency detection method for textured models that leverages multi-view information to achieve better performance. Specifically, we have proposed a texel descriptor that utilizes multi-scale operations to capture human eye interests. MVSM-Fusion integrates both results effectively and quantitative analysis demonstrates that the proposed method is effective.

Related Works

Methods using hand-crafted features. In the early days of saliency modeling for meshes, several methods adopted geometric features to estimate the visual importance of different parts of the mesh. For example, Lee, Varshney & Jacobs (2005) proposed a method that utilizes Gaussian-weighted curvatures at multiple scales, inspired by the center–surround mechanism of the human visual system. Song et al. (2014) proposed a method based on the log Laplacian spectrum that measures the difference between the original mesh and a simplified mesh to capture global features. Leifman, Shtrom & Tal (2016) designed a method that detects interesting surface regions by exploiting the observation that salient regions are distinct both locally and globally. Song et al. (2016) presented a local-to-global framework for mesh saliency computation, addressing the problems with a statistical Laplacian algorithm, pooling and global distinctness. Hu et al. (2020) represented the geometry of a mesh using a metric that globally encodes the shape distances between every pair of local regions and optimizes this problem to get mesh saliency.

Methods using learning features. Learning-based models generally have just a thin layer due to lack of training data. For example, Chen et al. (2012) built a M5P regression model based on hand-crafted features to predict Schelling points. Its training comes from a dataset containing 400 meshes spread evenly among 20 object categories. Wang et al. (2019) introduced a shallow convolutional neural network (CNN) that only contains five layers. Although these works display potential in saliency detection, the size of the dataset limits their performance. To avoid the limit from dataset, Song et al. (2021) introduced a GAN network which use multi-view projected images and natural images as the training data and its results perform better than state-of-the-art methods. To accomplish the same objective, Liu et al. (2022) proposed a novel attention-embedding strategy for 3D saliency estimation by directly applying the attention embedding scheme to 3D mesh.

Methods using meshes with properties. Several studies have investigated which regions of a textured model are more likely to attract human eye fixation. Yang, Wang & Li (2010) proposed a salient region detection method for 3D textured models that combines mesh and texture saliency through nonlinear normalization. Later, Yang et al. (2016) incorporated entropy factors to improve texel feature computation precision. More recently, Nouri, Charrier & Lézoray (2017) introduced a novel method called Global Visual Saliency, which uses Dempster-Shafer theory to generate saliency maps for colored meshes.

Methods in point clouds. Point clouds have become an increasingly popular representation of 3D objects due to their efficiency and accuracy. Shtrom, Leifman & Tal (2013) proposed a method based on simplified point feature histogram (SPFH) to compute the dissimilarity between two points and detect critical points in dense point clouds. Tasse, Kosinka & Dodgson (2015) introduced a cluster-based saliency method that evaluates the distinguishability and spatial distribution of each cluster. Zheng et al. (2019) showed a metric that updates saliency maps iteratively by gradient-based point dropping on the spherical coordinate system to obtain point cloud saliency. These methods can locate critical points that significantly influence mesh shape and geometry. Additionally, some methods based on salient or critical point theory have achieved impressive results, such as Leal et al. (2021) and Song et al. (2022).

Image saliency methods. Early image saliency models (Schölkopf, Platt & Hofmann, 2007; Achanta et al., 2008) were mostly focused on the bottom-up visual attention mechanism. Generally, these works assume that certain visual features of attractive regions always attract humans’ attention. On the basis of it, researchers created saliency maps that predict human gaze distribution using different visual features such as color, edge, and other special features in different domains. Recently, deep-learning-based statistical saliency models, such as Borji & Itti (2013) and Thomas (2016), have achieved significant improvements because of neural networks’ powerful learning capability and large-scale datasets. For example, Salicon-Net (Thomas, 2016) and Deep-Net (Pan et al., 2016) all adopted VGG16 network as the basis and made some changes on it to explore more effective network architectures to learn better features. Liu et al. (2018) proposed an architecture which adopts a multi-resolution convolutional neural network (Mr-CNN) to infer these three types of saliency cues from raw image data simultaneously. To date, more and more works are proposed.

Overview of the Eye Moving Collection

The study has been approved by the Institutional Review Board (IRB) of Computer Graphics and Simulation Lab in Changchun University of Science and Technology, with the approval number IRB-2021-0825.

Details and Calibrations

Apparatus: We use the HTC VIVE headset to present the 3D meshes in the virtual scene. The headset’s display resolution is 2, 448 × 2, 448 with a refresh rate of 90 Hz. Human eye fixation data is characterized by virtual scene coordinates. We use the Droolon F1 VR eye tracking accessory, embedded in the HTC VIVE headset, to capture human eye movement. The contents displayed in the helmet are shown in the Fig. 2. Furthermore, this module was capable of recording human eye movement within the field of view (FOV) with an error of less than 0.5° . When the accessory is active, the sampling rate is set to 90 Hz which is the same as the refresh rate. Each human eye movement recorded by an eye-tracking accessory is characterized by world coordinates (x, y, z) and classified as human eye fixation by Tobii software.

Figure 2 The images displayed on the screen in the HTC VIVE Pro2 are arranged with above order.

First calibration

Tobii provided a calibration to calibrate pupil distance and wearing height. After wearing the HTC VIVE Pro2 helmet, each observer needs to calibrate. Firstly, each observer is required to adjust the helmet position to make sure that human eyes are set in the right place. This is so that human eyes can be detected fully by the Droolon F1 device. The second step is for observers to rotate the switch to make the helmet pupil distance the same as their own. The last step is to reduce the inaccuracy between eye gaze and fixation computed by the device. A blue ball would be placed randomly on the screen, and observers should gaze at it before it disappears. The third step should be continued until the screen displays “The calibration is complete”. Besides, each observer in the experiment should remove their glasses, which greatly influences the accuracy of results.

Second calibration

It is necessary to perform this calibration step since there is always a deviation between the white ball and the purple ball in Fig. 3. In order to minimize this deviation, we designed this calibration in such a way. In detail, after the first calibration, observers would see one purple ball fixed in the center and one floating white ball reflecting their gaze in real time. While observers gaze at the purple ball, they could press W, A, S or D on the keyboard. This would make the white ball cover the purple ball as completely as possible. After the two balls are mostly aligned, the calibration ends, and each observer’s deviation configuration is saved for later data analysis. The projected images will display on the screen in turn, as shown in Fig. 2, once this calibration has been completed.

Figure 3 Screenshot of second calibration.

The purple ball is fixed at the center of the screen while the white ball that reflects the human eye movement is moving. When human gaze the purple ball, the observers could use W, A, S or D which represents UP, LEFT, DOWN or RIGHT respectively to move the white ball to near the purple ball.

Stimuli and observers

Image generation

As a result, we created 2D visual stimuli from a 3D model by creating images of (512 × 512) pixels as required by Unity’s material package, regardless of how many vertices the mesh contains. All of these images were rendered using OpenGL in perspective projection without any lighting shaders. Each textured model was centered within the virtual scene and aligned with the world’s coordinate when rendered in 2D. All images were taken with a camera oriented toward the center of the scene. We set up that each 3D object is upright oriented along the z-axis and represent it as a set of projected 2D images which are taken as input by our method. More specifically, in our experiment, we created n rendered views for a 3D model with the viewpoints subject to azimuth = i × 360/n, where i ∈ {1, 2, …, n}, where azimuths are measured in degree.

Configuration in unity

When testing the eye-tracking accessory, we found two problems. The first one is that it is unable to accurately reflect the gaze of the observer when the observer looks at the marginal region of the screen. Another one is that head movement greatly affects human eye fixation accuracy. In order to avoid these problems, each image occupies about half the screen of headset. In our experiment, the images are placed before the eye at a 5-unit distance in Unity.

Observers and other configurations

Everyone in the experiment sat in the same seat in the same environment. After the two calibrations, the human eye movement collection program ran and different images appeared on the screen (Fig. 2). In order to avoid the effect of similar images, we display images generated from different models in random order. The experiments were conducted in a free-watching setting, and no specific task was given to observers other than to observe the content of stimuli. A total of seven participants were involved in this experiment. They are aged between 22 and 30 and all have normal or almost normal vision. In order to avoid the effect of prior knowledge, nobody involved in the experiment is related to our research. Additionally, they have no clear understanding of the goals of our experiment. On average, it took 10 min for one observer to finish the experiment.

Data processing

We recorded each observer’s eye movement data in local files as soon as the string “Collection is complete!” appeared on the screen. The local file contains a sequence of fixations. Each fixation is characterized by a spatial 3D position (x, y, z) in Unity world coordinates. We set the display plane at a fixed position while the camera moves, so the z-component of each fixation corresponds to the distance between the camera and the plane.

However, datasets obtained directly without processing cannot be considered ground truth due to their noise sensitivity. Based on our observations, distraction is the main source of error, leading to outlier points, such as those shown in the first row of Fig. 4. To address this issue, we aim to identify and remove outlier points while preserving the original distribution of human eye fixations. Specifically, we group fixations into clusters using the K-means algorithm and remove clusters that contain scarce gaze data.

Figure 4 The first line is unprocessed eye movement data, and the second line is eye movement data processed by the data processing.

In details, each observer’s eye fixations are mapped to interface coordinate (x, y) which can be written as P(xi,yi). We denote that P = {P(x1,y1), P(x2,y2), …, P(xn,yn)} and n = |P| is the number of human eye fixations set. Then, all human eye fixations are split into m clusters and they are C = {C1, C2, …, Cm}. So, our target is to minimize the error ∑i=1m∑x∈Ci||x−μi||22, where μi=∑x∈Cix|Ci|. Our experimental setup initially involved setting m to 14, which was calculated by multiplying the number of observers by two. This resulted in obtaining a series of classified human eye fixations. Given that our objective was to eliminate outlier human eye fixations, we conducted a count of the number of fixations in each cluster and sorted these clusters in descending order based on their size. Ultimately, we removed all the human eye fixations in the last seven clusters. As shown in Fig. 4, the processed results exhibit a clear improvement compared to the raw results, with the disappearance of outlier points, reduced density of points along the model edge, and other notable differences.

Method

In this experiment, informed consent was obtained from each research participant and was documented in written form. In this section, we describe the details of MVSM-Fusion and Fig. 5 shows the pipeline of the proposed method.

Figure 5 MVSM-Fusion pipeline.

Overview of method

From Fig. 5, we can see that the MVSM-Fusion method involves two components: DV-Fusion and MSGI. For DV-Fusion, it requires three steps. Firstly, four 2D images are generated by projecting the given textured model onto specific view positions, and they are taken as input. Secondly, the 2D image saliency detection method is utilized, as shown in Fig. 6. Thirdly, Unified Operation is used to fuse multi-view saliency maps. For MSGI, it requires two steps. At the beginning, given a textured model, we compute each vertex’s texel descriptor. Secondly, we calculate the geometry saliency of a textured model via the difference between multi-scale saliency maps.

Figure 6 Comparisons of DV-Fusion based on different 2D saliency detection method.

From left to right: AC (Achanta et al., 2008), Deep-Net (Pan et al., 2016), FT (Achanta et al., 2009), GBVS (Schölkopf, Platt & Hofmann, 2007), LC (Zhai & Shah, 2006), Salicon (Thomas, 2016), Simple-Net (Reddy et al., 2020) and multi-view images. From top to bottom, “+X”, “+Y”, “-X” and “-Y” represents different views in local mesh coordinate.

DV-Fusion

Our approach to designing the DV-Fusion algorithm began with a conjecture: an optimal 3D saliency map corresponding to a given texture model could be derived. We further postulated that the 2D saliency maps in each view image represented the projection of the optimal 3D saliency map onto 2D space. This contained partial information about the optimal map. Fusing information from multiple views of 2D saliency maps is necessary to obtain a better 3D saliency map.

Thus, integrating saliency maps from different views into the same projection space posed a challenge. Through back-projection of the 2D saliency maps onto the 3D mesh, we observed that the histogram distributions of overlapping regions in the projected images of two views were highly similar, as illustrated in Fig. 7B. Based on this finding, we hypothesized that 2D saliency maps were linearly related. For each patch in overlapping parts, we divide the saliency values by another one. Additionally, we noted that the saliency values in the overlapping regions approximated a Gaussian distribution as observed from the histogram in Fig. 7C. To capture this distribution more accurately, we employed a mean-of-values representation and developed the Unified Operation.

Figure 7 The reason for Unified Operation.

(A) Shows the results that the image saliency of the two views is back-projected to the mesh. (B) Shows the histograms, which from two views, of overlapping part. (C) Shows the saliency histogram of the overlapping part after divide operation.

We design the DV-Fusion method as follows:

1. Given a set of patches P = {p1, p2, …, pn}, where n is the number of triangles in the textured model, we generate four images at {(d, 0, 0), (0, d, 0), (−d, 0, 0), (0,  − d, 0)} with projection operation, where d is the distance between the coordinate origin and camera position in local mesh coordinate. Then we get four images {I1, I2, I3, I4}.

2. We use the 2D saliency detection method on Ij and get one image saliency map called ISj, where j ∈ {1, 2, 3, 4}.

3. Inversely projecting each pixel in ISj to the corresponding patch in the textured model. Then we get one patch set and we call this set as Pj=p1j,p2j,…,pnj.

4. We define the overlapping part between Pj+1 and Pj,j+1 as Pj,j+1 = Pj∩Pj+1 and the number of Pj,j+1 is m. For each item Pij,j+1 in Pj,j+1, calculate its rate factor fij,j+1=puj+1/pkj, where u, k is the index of Pij,j+1 in pj, pj+1 respectively. Then we get one sets Fj,j+1=f1j,j+1,f1j,j+1,…,fmj,j+1. We define transformation factor fj,j+1 is the mean of three most frequent value in the histogram of Fj,j+1.

5. Finally, we use transformation factor fj,j+1 to unify Pj and Pj+1, like Pj+1 = Pj+1/fj,j+1.

6. Repeating the (3)–(5) operation until j = 3.

One of the advantages of the DV-Fusion framework is its flexibility in accommodating various 2D saliency detection methods. To identify the most appropriate method, we compared several techniques, including traditional and learning-driven algorithms, and evaluated their performance in terms of visual quality and quantitative metrics, as shown in Fig. 6 and Table 1. Our findings reveal that the AC-based, FT-based, and LC-based DV-Fusion methods generate saliency maps that closely resemble the ground truth distribution. Among the non-learning methods, the GBVS-based approach exhibits superior performance, although the salient and non-salient regions’ boundaries are less distinct. Notably, the Simple-Net-based DV-Fusion achieves the most effective visual performance, as well as the highest values in the evaluation metrics, including CC, SIM, NSS, and EMD (in Table 1). Consequently, we selected Simple-Net as the 2D saliency detection method to compute the saliency map for DV-Fusion, denoted as SMFusion.

Table 1 Comparisons between DV-Fusion based on different 2D saliency detection methods including traditional and learning works.

	Methods	SIM ↑	CC ↑	NSS ↑	EMD ↓	
Baseline	Yang et al. (2016)	0.36	0.41	0.95	4.33	
Traditional method	AC	0.49	0.50	1.03	3.86	
	FT	0.49	0.49	1.11	3.80	
	GBVS	0.50	0.55	1.34	3.54	
	LC	0.49	0.52	1.17	3.69	
Learning method	Deep-net	0.53	0.59	2.25	2.04	
	Salicon	0.51	0.54	2.08	2.48	
	Simple-net	0.58	0.70	2.35	1.91	

To confirm the effectiveness of DV-Fusion, we compared its results with those of the baseline method, which employs an outdated 2D saliency detection algorithm. Our experiments demonstrate that even with old 2D saliency detection methods, the framework still outperforms the Yang method. Therefore, our observations and comparisons attest to DV-Fusion’s superior performance over the Yang approach (Yang et al., 2016).

Multi-scales geometric information

In contrast to mesh models, texture can induce perceived significance even when geometric insignificance exists. To address this issue, we adopted the idea proposed in Yang et al. (2016), which combines texture and geometry features. Specifically, we focused on local colorful differences as a representation for texture relevance (Yang et al., 2016) and leveraged the salient point detection methods used in point-cloud analysis (Tasse, Kosinka & Dodgson, 2015) to develop a texel descriptor that captures the two characteristics of human eye perception: (1) sensitivity to convex polygons and (2) preference for regions with prominent texture changes, as shown in Fig. 8. And we can find that the value of fi decreases as the polygon flattens, and the weight of each vertex in the polygon becomes more uniform.

Figure 8 The vi is the core vertex and vb represents the calculated vertex position.

Although the DV-Fusion approach has the advantage of adaptively integrating the saliency maps computed by different 2D saliency detection methods, it has a limitation that saliency values in certain regions, such as the top or bottom of an object, may be incorrect because they are beyond the typical range of human eye perception. To address this issue, we introduced the MSGI method, which covers each vertex’s saliency value and thus avoids the patch disconnection problem in DV-Fusion. The MSGI method satisfies the requirement of capturing both local and global features (Liu et al., 2016), because it encodes both the local texel descriptors and the geometric connectivity information across the mesh surface. The MSGI method details are summarized as follows:

1) Given a set of vertices V = {v1, v2, …, vn} and a set of colors C = {c1, c2, …, cn}, where n = |V| = |C| is the number of mesh vertex and ci is the color of vi.

2) We compute T = {t1, t2, …, tn}, where ti = (0.30 × r + 0.59 × g + 0.11 × b)/255. As shown in Fig. 8, we define texel feature fi for a vertex vi as: (1) fi=vi− ∑j=1|Φ|tjvj|Φ|,

where Φ is a set that contains one-ring vertices of the vi and |Φ| is the number of this set.

3) Just like (Lee, Varshney & Jacobs, 2005) and other methods based on multi-scale operation, we detect saliency value for each vertex in scale s ∈ {1, 2, 3, 4}: (2) Fis=∑j=1|ϕt|e−||vi−vj||fi ∑j=1|ϕt|e−||vi−vj||,

where ϕs is a set that contains s-ring vertices of vi and |ϕs| is the number of this set.

4) Saliency difference between scale s and s + 1 of each vertex dis,s+1 is calculated as: (3) dis,s+1=Fis,s+1−Fis,s+1.

We define Ds,s+1=d1s,s+1,d2s,s+1,…,dns,s+1 and normalize it, called SMMSGI.

5) Finally, for the saliency map SMend = α × SMFusion + β × SMMSGI, where α = 0.68 and β = 0.32 in our experiment.

Experiments analysis

In this section, DV-Fusion is implemented on a 4-view basis and the parameter of SMend is mentioned in previous section. All experiments were computed on a computer with an Intel Xeon(R) Gold 5120T CPU and a NVIDIA Quadro P5000 GPU. For the Yang method (Yang et al., 2016), we use our own implementations.

Deeping in DV-fusion

The mathematical notation used in this section is the same as in the previous section. In the experiment, we discovered that Fj,j+1 could be approximated with several values, so we designed the Unified Operation. In detail, the textured model is projected at 20 viewpoint positions, and then the sets F = {F1,2, F2,3, …, F19,20} are obtained. We choose F1,2 and compute a histogram based on its frequency, as shown in Fig. 7. We found that the sum of the three most frequent segments occupies about 75.8% of the histogram. This proportion is higher for other items in F, and the average proportion is 80.5%. Thus, to unify P1 and P2 , we use the mean of the three most frequent values, called transformation factor f1,2. Then, we use this transformation factor to keep P1 and P2 in one space. Repeating the previous steps in other items in F and different view saliency maps is unified. Finally, the obtained similarity scores between DV-Fusion and human eye ground truth demonstrated Unified Operation’s effectiveness.

We selected the 4-view-based-DV-Fusion method because it strikes the right balance between computational efficiency and saliency map quality. As previously stated, we assume that each single-view saliency map is represented in a projection space. While it is in principle desirable to project the 3D saliency map multiple times to increase accuracy, there are practical limits to the number of projections we can realistically perform. Therefore, we aim to identify a threshold that yields acceptable errors relative to the ideal result. To gauge this error, we computed similarity scores between tthe 3D saliency map projections and the ground truth human eye fixations.

Based on the results summarized in Table 2, we observe that the similarity scores improve as the number of projections increases in terms of both the CC and SIM metrics. However, from Table 3, we also note that the improvement in CC and SIM scores is modest, while the running time grows exponentially with the number of views. Hence, our conclusion is that the 4-view-based-DV-Fusion method provides a good trade-off between computational efficiency and accuracy, demonstrating robust performance on evaluation metrics despite using a relatively small number of projections.

Table 2 Similarity between positive X views projection and ground truth.

N-views	20-views	12-views	8-views	6-views	4-views	
SIM	0.71	0.63	0.59	0.59	0.58	
CC	0.76	0.70	0.64	0.64	0.63	

Table 3 The running time of the proposed method based on different views.

N-views	20-views	12-views	8-views	6-views	4-views	
Time(s)	29.086	16.042	10.120	7.355	4.924	

We summarize our idea as follows. As a result of the observation that multi-view saliency maps could be approximately unified, we developed DV-Fusion. In addition, we use 4-views-based-DV-Fusion since more views bring more time cost and not much visual improvement.

MVSM-fusion performance

Figure 9 displays the ground truth human eye fixations collected by the Eye Moving Collection experiment (‘Overview of the eye moving collection’) for several textured models, along with the corresponding saliency maps computed by our method. We find that the saliency regions identified by our method are highly correlated with the ground truth human eye fixations. Additionally, we observed that viewers tend to focus on the head area of textured models as this area is often color-coded red, representing saliency.

Figure 9 A gallery of textured model saliency detected by MVSM-Fusion is in the top part (A), and the ground truth eye fixations computed by our eye moving collection experiment are shown in the bottom part (B).

In contrast, Fig. 10A reveals that the Yang method (Yang et al., 2016) detects large “blob-like” areas, which generally correspond to ground truth data. However, for the Buddha model, some unsalient areas with high geometric importance are detected in visual perception. MVSM-Fusion outperforms the Yang method in this regard.

Figure 10 Comparisons of textured model saliency detected by different methods.

From left to right: Yang’s method (Yang et al., 2016), our method, Simple-Net (Reddy et al., 2020) and ground truth eye fixations maps computed by our experiment. For the output of Simple-Net, we inversely project 2D saliency maps to 3D mesh via 3D-Picking technology.

Figure 10B illustrates the inverse projection of Simple-Net’s output into a 3D mesh to compare the visual differences between the 2D saliency detection method and our proposed method. As evidenced by the black segments in (Figs. 10B, 10C, and 10D), they are distinct from one another. Moreover, of all the methods featured in Fig. 10, only MVSM-Fusion recognizes this difference as salient. Specifically, MVSM-Fusion highlights Buddha’s face and the left side flower of the Bunny, which other methods ignore in the current view due to certain parameters in the projection, such as distance, camera position, among others.

In contrast, MVSM-Fusion can fuse multiple salient features from different views to obtain the salient features of the textured model as a whole, which yields superior saliency maps that are highly consistent with the ground truth. The output from the Yang method (Yang et al., 2016) and MVSM-Fusion method consists of a 3D saliency map, whereas the output from Simple-Net results in a 2D image saliency map. Furthermore, no suitable similarity metric exists to compare two 3D saliency maps. To evaluate the reliability of the 3D saliency approach, we projected the saliency map from multiple views. We employed CC, SIM, NSS, and EMD to quantify the similarity between the saliency map projected by the proposed method and the ground truth human fixation in several views. When the CC, SIM, and NSS are larger, it indicates a higher degree of similarity. For EMD, a smaller value indicates greater similarity. Finally, the average value was calculated to determine the performance of the 3D saliency map, as presented in Table 4.

Table 4 The running time of the proposed method based on different views.

Similarty	SIM ↑	CC ↑	NSS ↑	EMD ↓	
Yang et al. (2016)	0.36	0.41	0.95	4.33	
Simple-Net	0.58	0.70	2.35	1.91	
MVSM-Fusion	0.60	0.72	2.44	1.75	

Table 4 clearly indicates that MVSM-Fusion outperforms Yang’s method (Yang et al., 2016) in terms of CC, SIM, EMD, and NSS. In addition, our dataset demonstrated that our method outperformed Simple-Net, as observed in Table 4. Our method may be superior for the following reasons. Firstly, many studies use geometric descriptors to calculate mesh saliency, which often results in localized results that fail to capture the overall characteristics of the mesh. This problem is evident in Fig. 10A, where a small yellow area is depicted. In contrast, MSGI can extract features from a larger area, as observed in Fig. 11D. Secondly, while 2D saliency detection algorithms have shown promising results on a single view, they lack information from other views. This makes multi-view fusion a viable and effective solution.

Is texture-based method enough for textured model saliency detection?

In this section, to explore MVSM-Fusion deeply, we conduct extra studies: we compare the performance of DV-Fusion, MSGI operation and MVSM-Fusion to find the quantitative role of MVSM-Fusion and its several parts, as shown in Fig. 11.

Figure 11 Comparisons between Yang method, DV-Fusion method, and MSGI method from a specific viewpoint.

Table 5 presents the quantitative results, which show that MSGI and DV-Fusion outperform Yang in several metrics. Notably, Yang and MSGI exhibit poor performance, as they prioritize geometry information over texture data. However, as texture information is more critical for detecting salient regions in textured models, this phenomenon is not surprising. Additionally, DV-Fusion showed superior performance in several similarity metrics after implementing MSGI, suggesting that MSGI captures information that texture-based methods might miss. This indicates that extracting both texture and geometry information is essential for accurately predicting salient regions.

Table 5 Comparisons between Yang, MVSM-Fusion and its two parts based on several metrics.

Similarity	SIM ↑	CC ↑	NSS ↑	EMD ↓	
Yang et al. (2016)	0.36	0.41	0.95	4.33	
MSGI	0.41	0.42	1.13	3.95	
DV-Fusion	0.55	0.60	1.88	2.21	
MVSM-Fusion	0.60	0.72	2.44	1.75	

Conclusion and Limitation

This article presents MVSM-Fusion, an effective method that combines texture and texel features for predicting the saliency of 3D textured models. Our experiments demonstrate the importance of texture features in saliency detection. In addition, we contribute a novel dataset of textured models annotated with human eye fixations.

Limitations and future directions. Despite its high performance, MVSM-Fusion has several problems. In certain viewpoints of the textured model, our method may split the model’s face into salient and non-salient regions, which is unrealistic. This issue could be due to the limited anti-noise capability of deep-learning-based approaches, as traditional saliency detection methods combined with DV-fusion do not exhibit this problem. In future work, we will aim to explore new strategies for developing 2D saliency detection methods that can be integrated into MVSM-Fusion. We will also investigate more advanced fusion techniques for combining multi-view saliency maps. Overall, we believe that our findings regarding textured model saliency will contribute to the advancement of research into human visual perception of 3D objects and other complex scenes.

Supplemental Information

Supplemental Information 1 Codes for DV-Fusion and MSGI

Click here for additional data file.

Supplemental Information 2 Human eye fixations

Click here for additional data file.

Additional Information and Declarations

Competing Interests

Author Contributions

Ethics

Data Availability

The authors declare there are no competing interests.

Ya Zhang conceived and designed the experiments, performed the experiments, analyzed the data, performed the computation work, prepared figures and/or tables, authored or reviewed drafts of the article, and approved the final draft.

Chunyi Chen conceived and designed the experiments, prepared figures and/or tables, authored or reviewed drafts of the article, and approved the final draft.

Xiaojuan Hu conceived and designed the experiments, prepared figures and/or tables, authored or reviewed drafts of the article, and approved the final draft.

Ling Li conceived and designed the experiments, performed the experiments, prepared figures and/or tables, authored or reviewed drafts of the article, and approved the final draft.

Hailan Li performed the computation work, authored or reviewed drafts of the article, and approved the final draft.

The following information was supplied relating to ethical approvals (i.e., approving body and any reference numbers):

This study was approved by the Institutional Review Board (IRB) of Computer Graphics and Simulation Lab in Changchun University of Science and Technology.

The following information was supplied regarding data availability:

The data is available at Zenodo: zhang. (2023). human eye fixations on some textured model (partly). https://doi.org/10.5281/zenodo.8131602.

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
