# Peer review of "Saliency detection of textured 3D models based on multi-view information and texel descriptor"

_PeerJ Computer Science, doi:10.7717/peerj-cs.1584_

## Round 0.1 · original submission · Major Revisions

Authors are advised to carefully address all the comments and concerns received from the reviewers. The article requires mentioned amendments before proceeding further.

·

Basic reporting

In this work authors proposed a an approach which captures the texture and geometry saliency, further it fuses texture feature and texel features. The method, called Multi-View-Saliency-Map Fusion (MVSM-Fusion).

Experimental design

Authors performed basic level of experimental analysis to prove the validity of their approach. Their approach is not compared with state-of-the-art approaches.

Validity of the findings

The performance of their method (MVSM-Fusion) is compared with Yang, DV-Fusion, MSGI operation and Simple-Net using the metrics viz. Pearson’s Correlation Coefficient (CC), Similarity Metric (SIM), Earth Mover’s Distance (EMD) and Normalized Scanpath Saliency (NSS).

Additional comments

The scientific writing style is missing in many places in this paper, authors needs to follow standard writing style to confirm the good flow of this article.
Figure.1 needs to come after it is described in text paragraph.
What is inferred from "Related works" section?
Put numbering to all equations.
Most of the references are old, only few references are from recent years.

Reviewer 2 ·

Basic reporting

The authors of the paper Saliency detection of textured 3D models 2 based on multi-view information and texel 3 descriptor
- Proposed and built a dataset via an experiment that records the human gaze on a textured model. Where seven observers were involved in their experiment, and the textured models were adopted from an existing study.
- Suggested developing a capable of reflecting human eye fixations on textured models to overcome the problem in a good manner. The study will contribute positively to the field.

Experimental design

- The present work focuses on a) building a dataset of 3D textured models with mapped human eye fixations. b) Introduce one saliency detection of textured model based on multi-view information and texel descriptor and make a quantitative analysis of it, showing that the proposed method is efficient.

- The study will be useful for scientists, academicians, biotechnologists, and environmentalists along with research scholars doing research in various sectors

- Introduction improvement: We suggest that authors should provide more detail relating to the study. That could include before “Our contribution is twofold: 1) We build a dataset of 3D textured models with mapped human eye fixations. 2)” lines 52 and 53.

- Language Improvement: We encourage Authors to read the work and correct some spelling mistakes to improve their work. For example, lines 24-25.

Validity of the findings

The authors’ approach suited the findings by employing the stated methods as follows: DV-Fusion and MSGI. For DV-Fusion, it requires three steps. Firstly, four 2D images are generated by projecting the given textured model onto specific view positions, and they are taken as input. Secondly, the 2D image saliency detection method is utilized. Thirdly, Unified Operation is used to fuse multi-view saliency maps. For MSGI, it requires two steps. In the beginning, given a textured model, they compute each vertex’s texel descriptor. Secondly, calculate the geometry saliency of a textured model via the difference between multi-scale saliency maps. Additionally, single out Simple-Net among others as the method for 2D saliency detection based on the results of observations and comparisons.

Therefore, employing this approach of methods followed by the authors will surely enhance of reflecting human eye fixations on textured models and be useful for the researchers in the field.

Results and Discussion:
The authors concluded according to their findings that the method outperforms the state-of-the-art method in terms of Pearson’s Correlation Coefficient (CC), Similarity Metric (SIM), Earth Mover’s Distance (EMD), and Normalized Scanpath Saliency (NSS). However, please specify the (EMD) which seems to be the lowest value in your findings.

General comments:
Look forward to the revised manuscript after the minor revision.

---

## Round 0.2 · accepted · Accept

Congratulations! Your research work is accepted for publication. Consider thee proofreading carefully.

·

Basic reporting

The article is well written.

Experimental design

Research questions are addressed in the revised manuscript.

Validity of the findings

Authors performed valid experimentation and support github link were provided.